# A revised Law Dome age model (LD2017) and implications for last glacial climate

Jason Roberts<sup>1,2</sup>, Andrew Moy<sup>1,2</sup>, Christopher Plummer<sup>2,3</sup>, Tas van Ommen<sup>1,2</sup>, Mark Curran<sup>1,2</sup>, Tessa Vance<sup>2</sup>, Samuel Poynter<sup>2</sup>, Yaping Liu<sup>4</sup>, Joel Pedro<sup>5</sup>, Adam Treverrow<sup>2</sup>, Carly Tozer<sup>2,6</sup>, Lenneke Jong<sup>2,3</sup>, Pippa Whitehouse<sup>7</sup>, Laetitia Loulergue<sup>8</sup>, Jerome Chappellaz<sup>8</sup>, Vin Morgan<sup>1,2</sup>, Renato Spahni<sup>9</sup>, Adrian Schilt<sup>9</sup>, Cecilia MacFarling Meure<sup>10</sup>, David Etheridge<sup>10</sup>, and Thomas Stocker<sup>9</sup> <sup>1</sup>Australian Antarctic Division, Kingston, Tasmania 7050, Australia

<sup>2</sup>Antarctic Climate & Ecosystems Cooperative Research Centre, University of Tasmania, Hobart, Tasmania 7001, Australia <sup>3</sup>Institute for Marine and Antarctic Studies, University of Tasmania, Hobart, Tasmania 7001, Australia

<sup>4</sup>State Key Laboratory of Cryospheric Sciences, Northwest Institute of Eco-Environment and Resources, CAS, Lanzhou, 730000, China

<sup>5</sup>Center for Ice and Climate, Niels Bohr Institute, University of Copenhagen, Copenhagen 2100, Denmark

<sup>6</sup>Faculty of Science & Information Technology, University of Newcastle, Callaghan, New South Wales 2308, Australia <sup>7</sup>Department of Geography, Durham University, Durham, DH1 3LE, UK

<sup>8</sup>University Grenoble Alpes, CNRS, IRD, Grenoble INP, IGE, F-38000 Grenoble, France

<sup>9</sup>Climate and Environmental Physics, Physics Institute and Oeschger Centre for Climate Change Research, University of Bern, Sidlerstrasse 5, CH-3012 Bern, Switzerland

<sup>10</sup>Climate Science Centre, CSIRO Oceans and Atmosphere, Aspendale Victoria 3195 Australia

Correspondence to: Jason Roberts (Jason.Roberts@aad.gov.au)

**Abstract.** Here we present a revised Law Dome, Dome Summit South (DSS) ice core age model (denoted LD2017) that significantly improves the chronology over the last 88 ka. An ensemble approach was used, allowing for the computation of both a median age and associated uncertainty as a function of depth. The revised chronology incorporates extended continuous annual layer counting to 853 m using chemical species with seasonally-varying behaviours. The annual layer counted age at

- 853 m is 2332 years before 2000 (y b2k) with an error of +13/-7 y, i.e. 2345–2325 y b2k. Below this depth, non-linear interpolation between age ties using a probability density function for age/depth is used to constrain and model the age of the ice. The ice-based age ties below the annual layer counted section are based on matching volcanic event markers, methane (CH<sub>4</sub>) gas concentration, isotopic composition of ice ( $\delta^{18}$ O) and the Last Glacial Maximum (LGM) dust peak to other records. For consistency, the timescale used for all matching is the AICC2012 timescale (Veres et al., 2013). The first ice-based age tie is
- the base of the annual layer counting record (2332 y b2k) and the age ties from ~2400–4000 y b2k are volcanic synchronised ice-based age ties. The detection of abrupt changes in CH<sub>4</sub> gas concentrations within the DSS record provides further independent gas-based age ties, including the tightly constrained 8200 y b2k event. The improved age control between 9000 and 21000 y b2k is supplemented by CH<sub>4</sub> and δ<sup>18</sup>O ice measurements (Pedro et al., 2011). Over the period 16600 to 18600 y b2k large changes in dust concentration, matched to the EDC dust record, are used to constrain two ice-based age ties. Unlike
- previous studies, where the modelling was used to simultaneously infer both age and snow accumulation rate, we made an independent estimate of the snow accumulation rate, where required, for the use of gas based age ties.

# 1 Introduction

Law Dome is a small (~200 km diameter), independent ice cap situated on the coast of East Antarctica. The site has a maritime climate and is strongly influenced by easterly airflow produced by low-pressure systems centred around 65°S (Curran et al., 1998; Morgan et al., 1997). The primary ice core drilling site is Dome Summit South (DSS) located approximately
4.7 km SSW of the Law Dome Summit (66.7697°S, 112.8069°E) at an elevation of 1370 m. The site has a high, relatively temporally uniform accumulation (~0.7 m y<sup>-1</sup>ie, ice equivalent metres per year, based on an ice density of 917 kg m<sup>-3</sup>), low surface temperatures (mean annual temperature of -21.8 °C) which preclude summer melt and relatively moderate wind speed (8.3 m s<sup>-1</sup>) which results in clear annual layer definition (Morgan et al., 1997; Roberts et al., 2015).

The Law Dome DSS ice core record is primarily based on a single deep core, which is described in Morgan and van Ommen (1997). The main DSS core (DSS-main) is augmented in the upper portion by splicing three overlapping ice cores, namely DSS99, DSS97, and DSS1213 to DSS-main that cover the epochs 1841–1887, 1888–1988 and 1989–2012 CE (Roberts et al., 2015). The purpose of the additional cores is twofold: first to improve sampling and measurement integrity, and second to extend the record forward in time to 2012 CE (Plummer et al., 2012). The upper ~125 m of the original DSS-main ice core is affected adversely in places by melt-percolation in the thermally drilled firn portion, and by poor core quality through a region

where the main electromechanical drill was commissioned. Additionally, improved analytical controls provided higher quality water stable isotope and trace ion chemistry measurements on the DSS97 and other recent DSS ice cores than previous early shallow sections of DSS.

The previous Law Dome DSS age model (LD1) was derived from a combination of annual layer counting, stratigraphic markers and flow modelling techniques (van Ommen and Morgan, 1996, 1997; van Ommen et al., 2004). Measured annual

- layers provided flow-thinning data which constrain a model-fit of the ice flow. The layer-thickness constraints required knowledge of the long-term palaeo-accumulation, which, for the latter half of the Holocene up to the preindustrial, is likely to have been unaffected by major systematic variation (van Ommen et al., 2004). Additional constraints on this model, which do not depend so strongly on palaeo-accumulation, arise from various independently dated horizons in the lower ice-sheet. These include the use of air-composition measurements at DSS ( $\delta^{18}$ O of O<sub>2</sub> and methane concentrations) to tie deglacial changes
- (Morgan et al., 2002; Pedro et al., 2011) to GRIP and GICC05 chronology. The precision of the record through this period (~9–19 ky b2k) is discussed by Morgan et al. (2002) and is limited by uncertainties in the firn air-trapping and hence offset between ice and gas ages, which arise from uncertainties in temperature and accumulation.

Here we present the revised Law Dome DSS ice core age model that improves the chronology presented by Morgan et al. (2002) and van Ommen et al. (2004) using new data. The revised chronology incorporates the extended annual layer counting to

30 2332 years (Plummer et al., 2012; Roberts et al., 2015), and the improved age control between 9 and 21 ky that is supplemented by methane (CH<sub>4</sub>) and  $\delta^{18}$ O ice measurements (Pedro et al., 2011). Compared to LD1, the new age scale has an additional 1545 y of annual layer counting, nine additional age ties, age and depth uncertainties for each individual age tie (compared to the bulk age uncertainty of 200 y during the Holocene for LD1) that are propagated throughout the age model, revised depths for some age ties, and revised ages at the tie points. Overall, the difference in the ages at the LD1 age ties range between 11

and 2042 y. The maximum depth of the oldest age tie is clightly reduced in the current age model (1182.302 m compared to 1182.657 m) while the corresponding age is slightly increased (88012 y b2k compared to 87212 y b2k). The revised Law Dome DSS age model is denoted LD2017.

- The age-scale uses continuous annual layer counting and ice and gas age ties to other records. The age ties below the annual layer counted section are based on matching Law Dome volcanic event markers, methane gas concentration ( $CH_4$ ), isotopic 5 composition of ice ( $\delta^{18}$ O) and the Last Glacial Maximum (LGM) dust peak to other records. For consistency, the timescale used for all matching is the AICC2012 timescale (Veres et al., 2013), although gas-based age ties via direct ( $CH_4$ ) gas-matching (Morgan et al., 2002; Pedro et al., 2011) use the GICC05 timescale. The AICC2012 timescale is an optimised multi-parameter and multi-site chronology developed from four Antarctic ice core records and is virtually identical to the layered counted GICC05 timescale for the last 60.2 ky (Veres et al., 2013).
- 10

Unlike some previous studies, this model does not simultaneously infer ages and accumulation, as in essence this propagates all errors in the ice physics into estimates of the accumulation. Rather, we estimate the accumulation independently, where required for the use of gas based age ties.

- Age models that rely on linear interpolation between age ties suffer from several shortcomings. The layer thicknesses in the 15 upper part of the interval between age ties tend to be too thin (and hence ages change too quickly) and conversely the layer thicknesses are too thick in the lower part (and ages change too slowly). In addition, there is a discontinuity in layer thickness (and hence implied accumulation rates) at the age ties. The age model used here aims to minimise these discontinuities in the layer thicknesses at the age ties by using a piece-wise parabolic annual layer model. This is somewhat similar to the smoothest annual-layer model of Fudge et al. (2014), although we use updated age ties and also fully propagate uncertainties throughout
- our model and report both the median and standard deviation of age at depth. The age model at 1 m vertical resolution is given 20 as Supplementary Information.

#### 2 Hybrid Age Model

25

The Law Dome DSS ice core age model is derived from a combination of directly counted annual layers and age ties to other records. All depths are in actual metres (not ice equivalent metres) downward from the surface, and accumulation rates are in  $m v^{-1}$ ie.

Annual layer counting is continuous down to 852.668 m using chemical species with seasonally-varying behaviours. These include water stable isotopes ( $\delta^{18}$ O and  $\delta$ D), hydrogen peroxide (H<sub>2</sub>O<sub>2</sub>) and trace ion chemical species (non-sea-salt sulphate  $(nssSO_4^{2-})$  and sea salt species (Cl<sup>-</sup>, Na<sup>+</sup> and Mg<sup>2+</sup>) (Plummer et al., 2012). The annual layer counted age at 852.668 m is 2332 y b2k with an error of +13/-7 y, i.e. 2345-2325 y b2k. The age error reflects uncertainty in counting/data interpre-

30 tation errors. Below this depth, non-linear interpolation between age ties reflects errors in the interpolation between age ties with uncertainty in both depth and age. This is a more stochastic process and assumes a probability density function (pdf) for age/positions. For the section below the annual layer counting, all ages are relative to the AICC2012 time-scale so all uncertainties are relative to AICC2012 and are not absolute.

# 3 Age Ties

Tables 1 and 2 list the ice and gas-based age ties used to constrain the revised Law Dome DSS age scale. Details relating to these age ties are given below.

## 3.1 Ice-based age ties

- The first ice-based age tie is the base of the annual layer counting record that extends the annual layer counted record of Plummer et al. (2012) by 56.309 m (or 310 y). The continuous counting of annual layers down to 852.668 m is equivalent to 2332 y b2k. The age ties from 2.4 to 4 ky b2k are volcanic synchronised ice-based age ties. The DSS volcanic record (nssSO<sub>4</sub><sup>2-</sup>) are synchronised to the EPICA Dome C (EDC) dielectric profile (DEP) record (Parrenin et al., 2012) on the AICC2012 timescale. Volcanic events identified and synchronised are listed in Table 1. The reliable matching of volcanic
- events during the early Holocene was not possible due to the discontinuous DSS sulphate record resulting from poor core quality.

Over the period 16.6 to 18.6 ky b2k, large changes in dust concentration are used to constrain two ice-based age ties. The late glacial dust concentration record for DSS is synchronised to the highly resolved EDC dust record (Lambert et al., 2008) on the AICC2012 timescale. Previous DSS dust concentration age ties were matched to the dust concentration records of the Antarctic

ice cores from Byrd and Vostok, which were tied to the GRIP age scale (van Ommen et al., 2004). The final ice-based age ties involve the synchronisation of the DSS  $\delta^{18}$ O and EDC  $\delta^{18}$ O records (Stenni et al., 2010) on the AICC2012 timescale. Like van Ommen et al. (2004) tie points were selected across Antarctic Isotope Maximum (AIM) events that show good agreement between the isotopic records, and with a minimum number of ties selected.

#### 3.2 Gas age ties

- Abrupt changes in  $CH_4$  gas concentrations within the DSS record provides further independent gas-based age ties (Table 2 ). The first of these two events occurs at the 8.2 ky event where the DSS  $CH_4$  signal is tied to the GISP2  $CH_4$  record (Kobashi et al., 2007) on the GICC05 time scale. The DSS  $CH_4$  record has a mean age sampling resolution of 12 y between ~7.5 and 8.3 ky and is matched to the GISP2  $CH_4$  record so that the large change in  $CH_4$  and the duration of the  $CH_4$  trough, is synchronised to the GISP2  $CH_4$  during the 8.2 ky event. Further air-composition measurements ( $\delta^{18}O$  of  $O_2$  and  $CH_4$
- concentrations) are used to tie the DSS deglacial changes (Morgan et al., 2002; Pedro et al., 2011) to GISP2  $CH_4$  on the GICC05 timescale.

To convert the gas based age ties to equivalent ice-based age ties, we estimate the age offset ( $\Delta$ age) between the gas and icebased ages. This is a function of the density of the firn, which in turn requires estimates of the accumulation rate, temperature and impurity loading. We estimate the temperature from the water isotope data and the accumulation rate from upper and lower

bounds that are based on vapour saturation pressure and upper heat flux respectively.

**Table 1.** Ice-based age ties used in the LD2017 timescale

| ${\rm Depth}\ {\rm m}$ | Age y b2k            | Age uncertainty y |
|------------------------|----------------------|-------------------|
| 761.020                | 1856 <sup>a</sup>    | +4/-7             |
| 852.668                | 2332 <sup>a</sup>    | +15/-8            |
| 868.16                 | $2438.48^{b}$        | 20                |
| 949.32                 | 3374.83 <sup>b</sup> | 20                |
| 951.98                 | $3421.48^{b}$        | 20                |
| 952.6                  | 3431.33 <sup>b</sup> | 20                |
| 966.49                 | 3675.96 <sup>b</sup> | 20                |
| 970.31                 | $3735.52^{b}$        | 20                |
| 987.43                 | 4053.86 <sup>b</sup> | 20                |
| 1130.895               | 16630 <sup>c</sup>   | 328               |
| 1132.220               | $18618^{d}$          | 328               |
| 1156.082               | 48112 <sup>e</sup>   | 1000              |
| 1164.717               | $60312^{f}$          | 1000              |
| 1170.452               | 72212 <sup>g</sup>   | 2000              |
| 1172.792               | 75812 <sup>h</sup>   | 2000              |
| 1179.542               | $84012^{i}$          | 2000              |
| 1182.302               | $88012^{j}$          | 2000              |
|                        |                      |                   |

 $^{a}$  Annual layer counting

<sup>b</sup> Volcanic synchronisation to EDC(AICC2012)

 $^{c}\,$  Dust, trailing edge of decrease after LGM with

EDC(AICC2012)

 $^{d}$  Dust, LD LGM maximum match with EDC(AICC2012)

 $^e~\delta^{18}{\rm O}$  pick of minima between AIM12/AIM13 with EDC

(AICC2012)

 $^f~\delta^{18}{\rm O}$  pick of AIM17 match with EDC (AICC2012)

 $^g~\delta^{18}{\rm O}$  pick of AIM19 match with EDC (AICC2012)

 $^{h}~\delta^{18}{\rm O}$  pick of AIM20 match with EDC (AICC2012)

 $^i~\delta^{18}{\rm O}$  pick of AIM21 match with EDC (AICC2012)

 $^{j}~\delta^{18}{\rm O}$  pick of minima between AIM21/AIM22 in EDC @

ca 88 ky (on AICC2012 agescale)

Estimates of surface temperature from average  $\delta^{18}$ O values for  $\pm 30$  y either side of gas-based age tie points are made, assuming a modern surface temperature of -22 °C, a modern  $\delta^{18}$ O isotope value of 22 ‰ and a slope in the range 0.44– 0.7 ‰°C<sup>-1</sup>. This gives the estimates of surface temperature at the gas-based age ties shown in Table 2.

We generate a range of possible  $\Delta$ age based on several accumulation rate constraints (see below). We then turn this range of estimates into a single combined pdf for  $\Delta$ age. This method overcomes previous uncertainties in the air-trapping, which arise from uncertainties in temperature and accumulation (Morgan et al., 2002; Pedro et al., 2011).

5

**Table 2.** Gas-based age ties used in the LD2017 timescale, and associated properties. Values are based on an isotopic slope of  $0.44 \%^{\circ} C^{-1}$  with values in parenthesis for an isotopic slope of  $0.7 \%^{\circ} C^{-1}$ . Given the noisy nature of the Ca<sup>2+</sup> measurements, have used the median value from within 0.5 m of the age tie.

| Depth    | Age                | Age<br>uncertainty | Surface<br>temperature | Vapour<br>pressue | Geothermal<br>ICE-5G(VM2) | Geothermal<br>W12    | $Ca^{2+}$  | $\Delta$ age | $\Delta$ age<br>uncertainty |
|----------|--------------------|--------------------|------------------------|-------------------|---------------------------|----------------------|------------|--------------|-----------------------------|
|          |                    |                    |                        | accumulation      | accumulation              | accumulation         | 1          |              |                             |
| m        | y b2k              | У                  | °C                     | $m y^{-1}$ ie     | m y <sup>-1</sup> ie      | m y <sup>-1</sup> ie | $ngg^{-1}$ | У            | У                           |
| 1101.137 | 8178 <sup>a</sup>  | 30                 | -19.75(-20.59)         | 0.699(0.711)      | 0.228 (0.208)             | 0.220 (0.201)        | 3.34       | 107.8        | 50.0                        |
| 1101.772 | $8270^{a}$         | 30                 | -19.46(-20.40)         | 0.734(0.715)      | 0.238 (0.215)             | 0.227 (0.205)        | 3.83       | 104.1        | 48.3                        |
| 1108.640 | 9333 <sup>b</sup>  | 150                | -19.05(-20.14)         | 0.743(0.720)      | 0.253 (0.225)             | 0.238 (0.211)        | 1.72       | 103.0        | 46.9                        |
| 1121.290 | 11681 <sup>c</sup> | 75                 | -22.28(-22.17)         | 0.676(0.678)      | 0.194 (0.196)             | 0.196 (0.199)        | 2.59       | 121.8        | 57.8                        |
| 1125.190 | $12819^{d}$        | 102                | -26.64(-24.92)         | 0.595(0.626)      | 0.134 (0.154)             | 0.147 (0.169)        | 5.19       | 156.8        | 83.8                        |
| 1129.040 | 14687 <sup>e</sup> | 30                 | -26.25(-24.67)         | 0.602(0.630)      | 0.141 (0.161)             | 0.172 (0.196)        | 4.21       | 147.7        | 75.6                        |
| 1138.482 | $28780^{f}$        | 74                 | -35.63(-30.57)         | 0.454(0.530)      | 0.078 (0.104)             | 0.091 (0.126)        | 15.63      | 254.4        | 162.4                       |
| 1144.280 | 35480 <sup>g</sup> | 184                | -34.21(-29.68)         | 0.474(0.543)      | 0.083 (0.109)             | 0.097 (0.130)        | 5.88       | 244.9        | 152.0                       |
| 1146.700 | $38220^{h}$        | 197                | -32.34(-28.50)         | 0.501(0.563)      | 0.091 (0.119)             | 0.108 (0.141)        | 14.57      | 211.9        | 128.7                       |

a 8.2ky bracket

 $^c~\mathrm{CH_4}$  YD to Holocene transition (GICC05)

 $^{d}$  CH<sub>4</sub> BA to YD transition (GICC05)

 $^{e}$  CH<sub>4</sub> Older Dryas to BA transition (GICC05)

 $^{f}$  CH<sub>4</sub> DO4 transition (GICC05)

 $^{g}$  CH<sub>4</sub> DO7 transition (GICC05)

<sup>h</sup> CH<sub>4</sub> DO8 transition (GICC05)

# 3.3 Accumulation constraints

To estimate the pdf for the  $\Delta$ age offset for gas-based age ties, estimates of palaeo-accumulation are required. To allow for uncertainties to propagate fully through the age model, we use two estimates of accumulation, providing a lower and upper bound on likely accumulation rates.

<sup>5</sup> We estimate an upper bound on the accumulation rate (Table 2) using the method of Jouzel et al. (2003) to account for changes in the atmospheric saturation vapour pressure. Specifically, the modern accumulation is scaled by the ratio of the temperature derivative of the saturation vapour pressure evaluated at the historical conditions compared to modern conditions. We use the saturation vapour pressure formulation of Murphy and Koop (2005) which is accurate to within 0.025% for temperatures greater than -163 ° C. The strength of the atmospheric temperature inversion ( $T_{inv}$  in ° C) is

10 
$$T_{inv} = T_{surf} + 1.0 + 0.67 (T_{surf} - T_{surf,modern}),$$

(1)

 $<sup>^{</sup>b}$   $\delta^{18}$ O air modern end transition hump (on GICC05)

where  $T_{surf}$  is the estimated surface temperature (° C) at the time under consideration,  $T_{surf,modern}$  the modern surface temperature (° C), the factor of 0.67 is the ratio of changes in inversion strength to surface temperature (Jouzel et al., 1987) and the offset of 1.0 ° C is the estimated modern inversion strength (van Ommen et al., 2004).

The geothermal heat flux at the site provides an lower bound on accumulation rate (Table 2). Specifically, an accumulation rate below this lower bound would result in insufficient advection of cold from the surface to counteract the geothermal heat flux, leading to basal melting, which can be seen not to have occured in the past, as relict ice from the last interglacial or earlier remains at the base of the icesheet (Morgan et al., 1997).

We assume that the temperature ( $T_B$  in °C) at the base of the ice sheet is at the (ice thickness dependant) melting point, and the surface ( $T_S$  in °C) at the temperature estimated from the water isotope record (Table 2). We then calculate the accumulation

- rate required to maintain this temperature difference. Specifically, using ice sheet thickness (z in m) from two independent reconstructions, ICE-5G(VM2) (Peltier, 2004) and W12 (Whitehouse et al., 2012a), the basal temperature (melting point) is taken as  $T_B = 0.01 - 8.7 \times 10^{-4} z$  (Paterson, 1994, Chapter 10). This excludes the influence of impurities, which tend to depress the melting point and hence the accumulation values would need to be larger to compensate for this. Therefore as a lower bound, excluding this effect is conservative.
- The temperature differential between the surface and base of the ice sheet can be estimated from the near divide steady state temperature equation (Paterson, 1994, pages 216–220), viz:

$$\theta_B = \left[\frac{\pi}{2\gamma} \operatorname{erf}\left(\frac{\gamma}{2}\right)\right]^{0.5} \tag{2}$$

where  $\theta_B = \frac{K(T_B - T_S)}{Gh}$  is the dimensionless temperature difference,  $\gamma = \frac{az}{\kappa}$  is the Peclet number, *a* is accumulation (my<sup>-1</sup>ie), *G* is geothermal heat flux (72mW m<sup>-2</sup>, van Ommen et al. 1999), thermal conductivity  $K = 9.828 \exp(-5.7 \times 10^{-3}(T + 273.15))$ (W m<sup>-1</sup>K<sup>-1</sup>),  $\kappa = \frac{K}{\rho C}$  is the thermal diffusivity (m<sup>2</sup>s<sup>-1</sup>), and the specific heat C = 152.5 + 7.122(T + 273.15) (J kg<sup>-1</sup>K<sup>-1</sup>).

As per Whitehouse et al. (2012b) we have assumed a constant ice sheet thickness between 30 and 20 ky, and a linear change between present ice sheet thickness at 100 ky and 30 ky.

To simplify the calculations we replace the error function with a rational function approximation (Abramowitz and Stegun, 1968) valid for  $x \ge 0$ ,

$$\operatorname{erf}(x) = 1 - (0.3480242y - 0.0958798y^2 + 0.7478556y^3) e^{-x^2},$$
 (3)

where  $y = \frac{1}{1+0.47047x}$  and evaluate for accumulation rate (a) to ensure basal temperature above the local melting point.

# 3.4 $\Delta$ age offset

20

30

The age offset for the gas-based age ties can be estimated using firn densification models. Again, to fully propagate uncertainties throughout the age model, we use firn densification models both with and without impurity effects to estimate the sensitivity of our results to these factors.

The  $\Delta$ age is calculated using the steady state Pimienta firn densification model (Barnola et al., 1991), which is known to work well (Ligtenberg et al., 2011), using the modification of Freitag et al. (2013) to account for the effects of surface impurity

content (specifically through the  $Ca^{2+}$  concentration, see Table 2), and a bubble close off at a firn density of 830 kg m<sup>-3</sup> (Freitag et al., 2013). Surface snow density is calculated using the formulation of Kaspers et al. (2004), allowing for an Antarctic slope correction as per Helsen et al. (2008) and a surface wind speed of 8.3 m s<sup>-1</sup> (Morgan et al., 1997).

In addition to the bubble close off depth (and associated firm age) calculated above, atmospheric gases diffuse in the firm.
Therefore, we need to correct Δage for this effect, which we do using the method of Spahni et al. (2003); Etheridge et al. (1998); Trudinger et al. (1997). This correction is subtracted from Δage, and the uncertainties combined assuming independence. We ignore the spread of air age in the ice for the uncertainty calculations.

Subtracting the gas diffusion ages from the firn cut-off age estimates gives an  $\Delta$ age estimate for how much older the ice is compared to the mean age of the air in the ice the same depth. The firn cut off estimate is the weighted average of the

10 lower and upper bound. The eight lower bound estimates are based on geothermal heat flux (two isotope slopes, two thickness reconstructions, and with and without the influence of  $Ca^{2+}$ ). The corresponding four upper bound estimates age based on the saturation vapour pressure (two isotope slopes, and with and without the influence of  $Ca^{2+}$ ). To minimise biasing towards the lower bound, these estimates were weighted by a factor of 0.5. The resulting  $\Delta$ age estimates and associated uncertainties are shown in Table 2 and the full list of age ties is given in Table 3.

## 15 3.5 Thickness data

To further constrain the age model and reduce age uncertainties (see Section 5) we included one spot annual layer thickness data measurement from Roberts (1999) of 0.046 m y<sup>-1</sup> at a depth of 1038.22 m based on 19 y of high resolution isotope measurements from the analysis core DSS 1091. (Note, the depth of this core has previously been revised to a slightly greater depth than reported in Roberts 1999).

The standard deviation associated with layer thickness measurements was estimated from the ratio of the mean to standard deviation in the Law Dome accumulation record for 19 y moving windows over the 2 ky accumulation record of Roberts et al. (2015).

#### 4 Age interpolation models

The method of van Ommen et al. (2004) used the vertical strain rate and velocity profile calculated from fitting a Dansgaard and Johnsen (1969) profile to age ties. In particular, vertical velocity (or annual layer thickness) offsets were introduced between ties to account for discrepancies in modelled ages within the inter-tie point zones. This results in step changes in annual accumulation what are physically unrealistic. Furthermore, the use of a vertical strain rate profile with a smooth first derivative is internally inconsistent, as any step change in accumulation should distort the vertical strain rate profile.

The method used here involved piece-wise fitting a continuous linear vertical velocity profile. Specifically, a piece-wise parabolic annual layer thickness model is used, which is equivalent to a piece-wise linear vertical strain rate, a piece-wise parabolic annual accumulation rate or any combination of the two. With such a scheme any change in vertical velocity (or equivalently annual accumulation rate) is therefore smooth. It is similar to the ALT model of Fudge et al. (2014) although with

5

**Table 3.** Age ties, uncertainties in depth and age are 2 standard deviations except for the upper two ages (which are 3 standard deviations).  $\Delta$ age uncertainties are one standard deviation.

| Depth m                | age y              | $\Delta$ age y     |
|------------------------|--------------------|--------------------|
| $761.020 {\pm} 0.010$  | 1856+4/-7          |                    |
| $852.668 {\pm} 0.010$  | 2332+15/-8         |                    |
| $868.16 {\pm} 0.010$   | $2438.48{\pm}20$   |                    |
| $949.32{\pm}0.010$     | $3374.83{\pm}20$   |                    |
| $951.98{\pm}0.010$     | $3421.48{\pm}20$   |                    |
| 952.6±0.010            | $3431.33{\pm}20$   |                    |
| $966.49 {\pm} 0.010$   | $3675.96{\pm}20$   |                    |
| 970.31±0.010           | $3735.52{\pm}20$   |                    |
| 987.43±0.010           | 4053.86±20         |                    |
| $1101.137{\pm}0.010$   | 8178±30            | $107.8 {\pm} 50.0$ |
| $1101.772 {\pm} 0.030$ | 8270±30            | 104.1±48.3         |
| $1108.640 {\pm} 0.100$ | 9333±150           | 103.0±46.9         |
| $1121.290{\pm}0.010$   | 11681±75           | 121.8±57.8         |
| $1125.190{\pm}0.100$   | $12819{\pm}102$    | 156.8±83.8         |
| $1129.040{\pm}0.100$   | $14687{\pm}30$     | 147.7±75.6         |
| $1130.895 {\pm} 0.010$ | $16630{\pm}328$    |                    |
| $1132.220{\pm}0.010$   | $18618{\pm}328$    |                    |
| $1138.482{\pm}0.010$   | $28780{\pm}74$     | 254.4±162.4        |
| $1144.280{\pm}0.010$   | $35480{\pm}184$    | 244.9±152.0        |
| $1146.700 {\pm} 0.010$ | 38220±197          | 211.9±128.7        |
| $1156.082{\pm}0.010$   | $48112{\pm}1000$   |                    |
| $1164.717 {\pm} 0.010$ | $60312{\pm}1000$   |                    |
| $1170.452{\pm}0.010$   | $72212{\pm}2000$   |                    |
| $1172.792{\pm}0.010$   | $75812{\pm}2000$   |                    |
| $1179.542{\pm}0.010$   | $84012 {\pm} 2000$ |                    |
| $1182.302{\pm}0.010$   | $88012{\pm}2000$   |                    |

greater annual layer thickness discontinuity, which we attempt to minimise. Due to the rapid change in the depth derivative of layer thicknesses around 8.2 ky, a piece-wise parabolic model was fitted to either side of the 8.2 ky event, and a linear annual layer thickness model was used to span the 8.2 ky event. The piece-wise parabolic model only approximately passes through the age ties (in a least squares sense), and a small linear correction is applied between age ties to ensure the model matches the target age ties.

For two ties at depth  $z_1$  and  $z_2$ , separated in time by  $\Delta t$  and with a local annual layer thickness  $L(z) = az^2 + bz + c$ , the age difference  $(\Delta t')$  between the ties is

$$\Delta t' = \int_{z_1}^{z_2} \frac{1}{az^2 + bz + c} dz,\tag{4}$$

where Δt ~ Δt' for a good estimate. The evaluation of the integral is given by Gradshteyn and Ryzhik (1980, Section 2.172).
The values of a, b and c are directly related to the estimated annual layer thicknesses at (the subset of) age ties. Particle swarm optimisation (Pedersen and Chipperfield, 2010; Roberts et al., 2013) was used to optimise the layer thicknesses and minimise the cost function,

$$CF = \sum_{2}^{n} \left( \left[ \Delta t - \Delta t' \right]^2 + \lambda \left[ \Delta correction \right]^2 \right), \tag{5}$$

where the summation starts at the second age tie, because all ages in this model are relative, and are taken with reference to the 10 upper age tie. A change in linear correction across an age tie ( $\Delta correction$ ) is added to ensure that the model passes exactly through the surrounding age ties. Changes in this correction result in discontinuities in the annual layer thickness model, which we minimise. The optimisation was robust to a wide range of the relative weighting parameter  $\lambda$  and a value of 25.0 was used.

To avoid over-fitting only a subset of age ties were used directly in the particle swarm optimisation, but all age ties had to be honoured through appropriate linear corrections. In particular, five age ties above the 8.2 ky event and seven below were used, with the choice of the number of segments based on a generalised cross-validation analysis (Friedman and Silverman, 1989).

Particle swarm optimisation may focus on a local optima and fail to find a global optimum. To alleviate this behaviour, the particle swarm optimisation was restarted 1,000 times with random initial conditions to ensure good coverage of the parameter space, and each restart randomly selected which age ties would be used to define the parabolic segments.

## 5 Ensemble models of age scale

- 20 To allow for the computation of not only the median age model consistent with the age ties, but to also quantify the uncertainty in the age model, an ensemble approach was used. Specifically, 10,000 realisations of possible age tie sets were randomly generated from the probability density functions associated with both the depth and age of each age tie. The upper two ties (annual layer counting), have non-symmetric errors, so skew normal distributions with zero median (see Figure 1) were generated using the method of Ashour and Abdel-hameed (2010). The 1856 b2k age tie required a skew parameter of 3.73 to reproduce
- 25 the 95% certainty interval of -7/+4 y. The corresponding data for the 2332 y b2k age tie was a skew parameter of 4.68 for the -8/+15 y interval. For all other ties, symmetric normal distributions with a 2 sigma error specified in Table 3 were used. The age and ∆age uncertainties were handled independently and then combined to produce a target age. Due to the relatively close spacing of the age ties relative to their corresponding uncertainty, some of the random ensemble members have the order of age ties swapped, and must rejected.
- The lack of a tie point between the depths of 987.43 and 1101.137 m results in excessive uncertainty in the age model in this region (approximately 5.5 times greater than the results presented here). To alleviate this, we include the layer thickness data

of Roberts (1999) to further constrain the model. Any particle swarm optimisation solution that resulted in a layer thickness at 1038.22 m outside 3 standard deviations of the measurement of Roberts (1999) was rejected.

An age model was g