# Peer review of "A revised Law Dome age model (LD2017) and implications for last glacial climate"

_Climate of the Past, 2017_

## Referee Comment (RC1) · Anonymous Referee #1 · 20 Sep 2017

The Law Dome DSS ice core is an important palaeoclimate archive and so its age model is important and the methodology and outcome of a new reference age model expected to last for many years deserves publication. Unfortunately this paper, while providing (as SI) the outcome of the age model, provides a very inadequate justification for the procedures behind the model. The purpose of such a paper has to be to fully justify the validity of the age model, and this requires providing information that allows the reader to judge that; this paper doesn't do that. I will go into more detail but among the issues are:

*The top part of the model relies on layer counting but NO examples of the annual cycles are shown to justify the low error assigned

* The next set of ties are volcanic ties to EDC, but again absolutely no records of

this are shown so the matching (which can be a very tricky business) has to be taken entirely on trust

* Similarly the dust match is not shown, denying the reader any chance to judge whether the match points are well-defined

* The isotope matches are also not shown. This case is even more severe because in work at other sites, authors have tried to avoid del18O matches because they assume climate synchoneity, the very issue that is often of value. While this may be unavoidable here, the fact that it is not even discussed is not acceptable

* Methane matches are again not shown, so again we cannot judge whether they are appropriate

* On the other hand a large amount of the paper is devoted to a rather strange calculation of delta-age, which thankfully is small but extremely poorly known

* The palaeoclimate implications of the new age model are not discussed, which must be the main scientific interest of the new age model

* I am willing to be corrected but to my knowledge many of the data underlying this work have not been published and certainly not been posted on databases. I can find at NOAA palaeoclimate the oxygen isotope data and very low-resolution (inadequate for this study) methane data, but not the chemistry for layer counting, sulfate data for volcanoes, dust or methane. Authors are required at least to state how the data can be accessed, and preferably should deposit data on a recognised database.

In summary this paper needs a huge amount of work. I was tempted to simply reject it, but do not want to send the message that an age model paper is not appropriate. However this is a very long way from being the right age model paper, and needs very very major revision.

Specific comments:

A picky comment but please regularise your use of age units, consulting https://www.climate-of-the-past.net/for_authors/manuscript_preparation.html (see abbreviations section). You should replace y with yr or a as appropriate.

Related to this it's a valid choice to use b2k, but slightly perverse when stating that ages are AICC2012 ages, since AICC2012 is bp. You need a sentence to explain to the unsuspecting reader so that they don't accidentally introduce 50 year errors. Finally in Fig 3 b2k not B2k for consistency.

Section 1, page 3, para 3. Somehow here and elsewhere the authors imply that using a model to simultaneously infer ages and accumulation is inferior. This may be the case at LD where the geography, and the fact that most of the years are in ice very near the base, means that the layer thinning is badly-behaved and cannot be treated properly (although I would like to be convinced that the uncertainty in later thinning will be so great that you cannot improve on the factor 3 uncertainty in acc rate you end up with from doing accumulation independently). However in other cases such as AICC2012, it is a clear advantage to use the two together because it means that the known (and in the case of most AICC sites until the lowest layers, well-behaved) physics of layer thinning and snow accumulation is used as a self-consistent constraint on the age. Given the incredibly unconstrained accumulation rate that is derived later it seems rather ridiculous to suggest that not deriving accumulation is somehow a virtue. This needs rewording to explain the particular circumstances of LD.

Overall methodology: the methodology used here is to take a small number (below the layer counting) of fixed ages and fitting a smoothed function through them. This is a reasonable approach, but given that everything is being keyed to AICC2012, I am wondering if the authors considered simply using the software used to make AICC2012 (or working with AICC2012 authors), and simply including LD in the framework. Perhaps the reason not to do that was that it would produce something slightly different – an absolute age model incorporating LD, whereas the current approach really produces a relative age model (LD keyed onto AICC2012). But anyway it would be worth

discussing the approaches and justifying the one taken right at the start of section 2.

Section 2 – you must show at least one example of the annual layer counting near the bottom of the layer counted section so that the reader can judge the validity of your uncertainty estimate (showing a section including one of the uncertain years would be the most informative). This has always been done in age model papers (eg Sigl et al 2016 for WD), and seems mandatory unless the layer counting is being published elsewhere simultaneously (there is no indication of that).

Section 3.1. Again I consider it mandatory to show the synchronisation of the volcanic layers in figure form so the reader can judge whether you have made unequivocal matches or not. Again there is good precedent for presenting such evidence eg in Fujita et al (DF vs EDC) and Parrenin et al 2012 (V vs EDC).

Section 3.1 again. You need to also show the dust record match LD vs EDC. The previous matching referred to is shown in van Ommen et al 2004 at very coarse resolution – one could not possibly tie two records together within 1000 years let alone the strangely precise 328 years from that. So we need to see the data lined up please. This turns out to be crucial in dating the start of Termination I discussed later for example.

3.1 again. For the isotopes there are two issues. Firstly we need to see the data. The Stenni reference given contains no LD data – it is van Ommen et al (2004) that should be cited here for the LD data. The resolution and sharpness of signals is clearly essential in assessing these ties and we need to see the data lined up with EDC, with the tie points drawn on.

In addition, you need to explain the compromise you are making by using del18O data matches. There is a good theoretical justification that single events (like volcanic eruptions) and signals with a common source (dust) must be synchronous, but there is no reason the slow climate signals have to be synchronous and indeed papers such as WD (2013) made important papers out of discussing a non-synchroneity. By using the climate signal to tie the records you are losing any chance to discuss phasing

between signals around Antarctica. I understand the need for this compromise in order to develop a useable age model, but it requires discussion.

Page 4, line 30. "upper heat flux" – do you mean "limits based on the heat flux"? As written it doesn't quite make sense.

Sections 3.3 and 3.4. An enormous effort goes into this calculation but in the end it gives you accumulation rates with an uncertainty by factor 3! And I am not convinced by the basis for the upper or lower limits. For the upper limits you do the thermodynamic calculation of accumulation based on isotope values. But while this has theoretical validity for the Antarctic Plateau, and therefore perhaps for LD when acc rates were much lower in the LGM, it has no real validity for a high accumulation situation with synoptic precipitation. It no doubt gives some empirical estimate but I am not sure it really gives an upper limit.

The lower limit based on heat flux is based on the idea that there has never been basal melting at LD. I understand the present-day basal T is -7 so more than likely this is true, but I don't see what the assertion is based on. If there was basal melting at, say, 40 ka, then all we know is that it didn't melt the ice from the LIG (as an example, there is basal melting at Dome C today but ice from 800 ka still exists at the bed, and an assumption of no basal melt would be wrong). If you can justify this calculation then there are also a couple of strange details: page 7, line 15, you assume steady state, but why should the ice be in steady state if the accumulation rate is changing so hugely? And line 22, I don't understand the phrase "linear change between present ice thickness at 100 ky and 30 ky" – please explain.

In the end I don't see how these calculations, based on uncertain assumptions, constrain the delta depth or delta age at all. I was then rather astonished to find that you have access to some 15N data that at least constrain delta-depth, and give a physically based estimate of accumulation rate that you choose to ignore (page 17). I don't have a great proposal as to how to constrain delta age but I don't think it should form such

a large part of the paper when the basis for it is so weak. Page 11, line 10 "this [independent acc rates] aids in reducing uncertainties". How? This is far from self-evident.

Section 6.3. Please show how the newly dated core lines up with other cores in comparison to the old (Pedro) line-up . How does this affect the conclusions of previous papers? This is surely the whole point of making a new age model, to learn something about climate dynamics. I note in passing that the 1000 year shift at 19ka mentioned in the text must be entirely controlled by the dust match there, emphasising how crucial it is to show this in a figure. As an example of climate issues that should be discussed does the onset of TI at LD now look more like WAIS Divide or is it still firmly like the East Antarctic records?

Page 16, line 15. You have misinterpreted something in Parrenin 2007 here. The ratio of LGM to Holocene accumulation at EDC is at least a factor 2 (as easily seen in his Figure 3). The value of 1.4 you cite seems to be the ratio by which he considers previous estimates were wrong. This then requires a rewrite of this section.

Page 17, line 9. It's not obvious how this argument about accumulation related to dust concentrations should work, please spell it out. (I know the argument but I doubt most readers will).

---

## Referee Comment (RC2) · Anonymous Referee #2 · 21 Sep 2017

Roberts and coauthors present a new timescale for the Law Dome ice core. The new chronology improves upon previous LD timescales by adding some new age constraints, using a piece-wise parabolic method for interpolation between age constraints, and performing new delta-age calculations. The new methodology remedies the interpolation issues in previous timescales that resulted from quasi-linear interpolation. There are also significant changes in the timing of climate features due to both the addition and removal of age constraints. The delta-age calculation is changed substantially due to a different approach of calculating the accumulation rate.

There is no question that the old Law Dome timescale needed updating and thus this work is a positive step forward. However, the paper has major flaws.

First, none of the data supporting the tie points is presented and no indication is given

that these data sets will become publicly available. There is, simply, no ability to assess the quality of age constraints. This is problematic in its own right, but it is doubly so considering the stated purpose of this manuscript is to correct the errors in the previous Law Dome timescales.

Second, the manuscript provides no clear rationale for why to use a piece-wise parabolic interpolation method. With AICC2012 (Bazin; Veres, 2014) and the smoothest accumulation/annual layer thickness (Fudge 2014), the rationale was clear. Like those methods or don't like those methods, at least the reader understood the philosophy behind them. The piece-wise parabolic interpolation used here feels like a knee-jerk reaction to be the LD1 timescale being criticized for quasi-linear interpolation. Further, the piece-wise parabolic method breaks down at the 8ka tie point, indicating that it is not well suited for the Law Dome ice core.

Third, the authors choose to tie LD2017 to the AICC2012 rather than WD2014. In fact, the authors do not mention the WAIS Divide ice core in the manuscript at all. For instance, the WD2014 timescale is an order of magnitude more precise than AICC2012 at 4ka (20year uncertainty at WDC vs. 200year uncertainty at EDC). Given that Law Dome volcanic events have previously been matched to WDC (Sigl et al., 2013), there is no reason not to continue to do so. But there are still other questions. If AICC2012 is going to be the reference chronology, why not actually use the AICC2012 methodology that is now publicly available as IceChrono on Github (https://github.com/parrenin/IceChrono). In terms of AICC2012, by not incorporating it into the framework, it misses the opportunity to use Law Dome (as the highest accumulation site) to actually improve AICC2012. I realize the AICC2012 developers may not be excited to revise AICC2012 based on Law Dome, but the authors would be in a much better position for this manuscript if they had done this exercise.

Fourth, the manuscript is full of inconsistencies. For instance, there is an entire section (6.5 and Figure 7) to the inapplicability of modeled velocity profiles. Yet to calculate the lower accumulation limit, the authors use a parameterization from Paterson (1994)

which relies on a linearly increasing vertical velocity profile (i.e. a Nye profile).

Overall, reviewing this manuscript took an extraordinary amount of time and effort because there are good insights followed by inconsistencies and unexplained decisions. Below I provide detailed questions and comments for the manuscript with my word count approaching the manuscript's.

Major Issues: Choice of Reference Timescale I described above why the choice of AICC2012 as their reference chronology is perplexing but it worth expanding on a little more. For the mid-Holocene time period, the WD2014 age uncertainty is an order of magnitude better. EDC, within the AICC2012 cores, is an especially bad choice since its ice ages during this period are found through a convoluted inverse procedure but are ultimately mostly decided by the GICC05 timescale through EDML methane links with a delta-calculation and then volcanic ties between EDML and EDC. Why not just tie directly to WAIS Divide which is annually layer counted itself and is of the same, if not better, accuracy to GICC05 in the first place. Further, the EDC and EDML timescales have already been volcanically matched to WAIS Divide (though the synchronization is not published yet, it has been discussed at meetings) such that a timescale tied to EDC/AICC2012 will have to be shifted within a year anyway.

Choice of Method for Timescale Roberts et al. develop a new method for interpolating and assigning uncertainty for a timescale. The first question that came to mind was: if they are synchronizing to AICC2012, why not use the DatIce/IceChrono methodology? IceChrono is freely available. Interestingly, the authors don't even discuss why they made the choice they did. The Datice/IceChrono methodologies certainly have their quirks, so I'm not necessarily against not using them, but that option should certainly be discussed.

The authors also reference the ALT method of Fudge et al. (2014) but don't articulate why their method is a better choice. Their language is even more confusing since it sounds like they are doing something similar, but acknowledging that is doesn't really

work as well: "It is similar to the ALT model of Fudge et al. (2014) although with greater annual layer thickness discontinuity, which we attempt to minimize".

In general, I found it surprising that the authors never justified why using a piece-wise linear vertical strain rate is appropriate. I also do not understand if the authors considered that the vertical velocity pattern can vary through time. They seem to only consider changes in vertical velocity with respect to depth such that their statement that a linear vertical strain rate is equivalent to a piece-wise parabolic annual accumulation rate is only true in an ice sheet where the vertical velocity is fixed. This is unlikely to be the case where the ice temperature and fabric will impact the vertical strain rate profile in addition to the open question of how often the ice recovered in the core would have experienced divide flow.

But this all comes back to that the Datice/IceChrono and ALT methodologies have clear motivation. The former is an attempt to combine all uncertainties in an inverse procedure to find the best possible solution. The latter seeks to minimize the variations in annual layer thickness such that any variations are robust so as not to introduce artifacts from interpolation. What is the motivation for piece-wise linear?

Interpolation The interpolation uncertainty is not assessed. The method, by essentially finding a smooth annual layer thickness, is not accounting for the random variations in accumulation that occur between age control points. How does this compare to the rate that the interpolation uncertainty increases at, as described found by Fudge et al. (2014)? Additionally, the authors have the opportunity do a similar assessment on the most recent ~2000 years of Law Dome (which is much more limited that the ~30,000 years of WAIS Divide), but choose to do so for only ~450 year interval. Why not assess their method for the full interval of annual layer counting with different choices of "arti-ficial" tie points? Particularly since the lowest section of the annual layer interpretation is likely to be most uncertain and most smooth.

Climate Interpretation The amount of "climate" analysis is in this manuscript is very

small considering that the word "climate" is in the title. Two areas in particular need more focus: 1) the change in accumulation at 8ka. This is mentioned but not illustrated. It is, arguably, the most interesting climate feature of the Law Dome ice core and yet after reading this paper I cannot decided if it is even real or just and artifact of poor timescale development. 2) A specific comparison of the revised timing of the water stable isotope record and the detection of change points. I should be able to open this paper and find a figure of the stable water isotopes for the past ∼25ka on LD1 and LD2017 with the change detection shifts clearly illustrated. There should also be a table of the tie point depths with both the LD1 and LD2017 ages (and uncertainties) clearly shown.

Line by line comments P1L4 – "seasonally-varying" does not need to be hyphenated and use some other word than "behaviors" – no need to anthropomorphize P1L5 – why use y b2k? Everyone except the Danes uses bp 1950, including AICC2012 P1L5 – be more specific than "non-linear interpolation" since the old Law Dome timescales were not technically linear interpolation either. P1L6 – "age/depth" is awkward and I don't understand what you are really trying to say P1L7 – why not just write out methane throughout the abstract? It's easier to read a word than a chemical formula P1L9 – AICC2012 uses bp1950 P1L16 – the abstract does not mention the delta-age calculation, which is a critical part of the new timescale. And also doesn't mention how you independently estimate the accumulation rate. This is worth specifying, because recent work has done this using fractionation of gases in the firn, which is not what you do.

P2L20-30 – I'm confused about what timescale you are improving? Are the Morgan 2002 and van Ommen 2004 distinct from the LD1 (Pedro 2011) timescales? P3L1 – You might also give this a percentage of age. Change an 80ka tie point 2ka is not big deal. Changing a 20ka by 1ka (which you do) is a huge deal. P3L1 – "slightly" P3L1 – sentence starting "The maximum" – how do change the depth of a tie point? Shouldn't the depth be fixed and only the age able to be changed? P3L8 – be careful here. Only the AICC2012 gas chronology is nearly identical to the GICC05 gas timescale. The AICC2012 ice timescales for all the cores have large uncertainties relative to the GICC05 annually interpreted ice ages because there are few direct ties between the Antarctic ice timescales and the Greenland ice timescales. P3L8-10 – I have addressed this above, but why link the timescale to AICC2012 rather than WD2014. For the past 31ka, which is most of the period of interest of LD2017, WD2014 is a vastly superior choice for reference chronology since it iss annually resolved, more accurate than GICC05 in the Holocene, has better resolved methane measurements, and volcanic events can be directly synchronized. The authors are surely aware than a synchronization of EDC and EDML to WDC has been completed if not quite published yet. P3L11 – "unlike some previous studies" - reference the studies and explain why this was a bad decision. P3L12 – why are you being so coy about how you estimate accumulation rates? Just tell us already P3L15 – this description could be improved because it only applies when the layer thickness is decreasing, which is more common than increasing layer thicknesses, but not universal. Also, be specific about which way ages change, i.e. they would increase too quickly. P3L17 – Why do you choose a piece-wise parabolic annual layer model? P3L19 – this reference to the ALT method of Fudge et al. 2014 is confusing since you don't introduce the ALT method here. Then you go on to explain the difference with the ALT method. This all just feels weird because Fudge et al. picked apart the LD1 timescale based on its quasi-linear interpolation yet you make no reference to this in the preceding sentences even though they make the exact same points. P3L26 – see above. Try "seasonal cycles" rather than "behaviors" P3L29 – delete "counting", the error comes entirely from interpretation, not counting P3L30-31 – I do not understand this sentence P3L31 – What does "This" refer to? This paragraph needs to be written to improve clarity. P3L32-33 – I am concerned about the use age uncertainties relative to AICC for a variety of reasons. First, the rest of paper does not make this obvious. This is such a major assumption that it needs to be highlight EACH AND EVERY time an uncertainty value is given. Second, the AICC uncertainties are large – the uncertainty for EDC at 4ka is 200 years – particularly

during the volcanic matching interval (∼2-4ka). Thus, choosing AICC as a reference chronology seems like a particularly bad choice. Further, I do not understand why the authors do not incorporate the full uncertainties rather than just relative. This is not explained or justified. P4L1 – No data of the age ties is presented. No sulphate. No methane. No d18O of O2. No d18O. This paper is specifically correcting inaccurate age markers from LD1. How can I evaluate the quality of the age ties if they are not presented? P4L21 – you are playing fast and loose with the timescales. Be specific about the conversion from the GISP2 gas timescale to the GISP2 ice timescale to the GICC05 timescale. The reader should have enough information to do it the same way. While the GISP2 ice and GICC05 timescales are well synchronized with volcanic events, the gas timescales have an added step of the delta-age calculation except as abrupt transitions. P4L21 – what do you mean by "the first of these two events". What is the second event? Table 1 – How do you get a 20year uncertainty on the volcanic ties? Either the ties are correct and the uncertainty is <1 year, or they are wrong and provide no information. Table 1 – 328 year uncertainty? Table 1- please make another column for all of the superscripted info P5L1 – why 30 years? This seems too small since the firn densification process takes longer than 60 years P5L2-3 – justify the range of slopes chosen. This seems too narrow to me given the uncertainties for a coastal site across the glacial-interglacial transition. P5L5 – why does "this method overcomes previous uncertainties"? Are the same uncertainties still there, just with different inputs from you? P5L5 – What firn model are you using to convert temperature and accumulation to firn density and delta-age? You will discuss this below, but you need to alert the reader to that fact. P6L5 – justify using the Jouzel et al. method. Consider than Roosevelt Island has a nearly identical surface temperature to Law Dome but receives less than half of the accumulation. Law Dome is the least likely site with a deep ice core to have accumulation controlled by the saturation vapor pressure. P6L9 – Inversion temperatures? This needs much more justification and explanation. What is the uncertainty on the inversion factor? But more broadly, is this really a useful way to be going? P7L7 – please provide more information on the basal ice. The Morgan

paper uses the phrase "initial results" and given than it's now 20 years later, an update is in order. The presence of previous interglacial ice is key to your argument, but the only thing I saw in the Morgan paper, which I had to track down a printed volume of Annals for, was a blurry plot with some near-Holocene d18O values. P7L15 – why are you using this Paterson approximation? It's based on a linear vertical velocity profile, which you explicitly say in this paper doesn't apply to Law Dome. Do the calculation right with a transient 1-D thermomechanical model. At least one of the 20 coauthors has experience with this, or contact any one of the numerous ice-core/ice-sheet modelers who could do this in a day. P7L19 – Why do you use the van Ommen value when the Dahl-Jensen 1999 value is better constrained? But the real question is, do you think either of these values is accurate or do they result from potentially errors in the assumption of vertical velocities? P7L21 – This sentence is very confusing. If I'm understanding Figure 6 properly, you are not actually running this in a transient manner, only at distinct ages. So why not just give us the ice thickness value at the times you decided to calculate this for? P7L19-21 – Why all of the specificity in the thermal values? Can't I just find these in Cuffey and Paterson? If these values are actually important, you need to explain why. P7L23-26 – Do I really need to know how you evaluated the erf function? Won't python just do this for me? P7L28 – I don't follow how including impurities in the firn densification results is "fully propagating uncertainties". Given that the inclusion of impurities is hotly debated and relies on rough parameterizations, this doesn't seem likely to yield full uncertainties. Further, there are more basic uncertainties in the firn models. P7L31 – I do not agree that the Pimienta model "is known to work well". There is a lot of debate about appropriate firn models. I think most unaffiliated firn model users would choose Herron and Langway before the Pimienta model. You need to justify your choice firn model with more than a single reference. I would recommend looking at the Firn Model Intercomparison (Lundin et al., 2017, J. Glac.). Better yet, use the community firn model to calculate a spread of values. P8L1 – what's the sensitivity to the close-off density? 820 seems like another common choice for the close off density. P8L7 – what is a "firn cut-off age"? P8L9 – I don't understand what you mean by

"the firn cut-off estimate is the weighted average of the lower and upper bound"? What bounds are you talking about? P8L14 – Why have you not measured any d15N of N2? This is a powerful constraint on firn densification and would provide hugely useful information for this analysis. P8L15 – Rename the section title so it doesn't sound like ice thickness. Maybe "select annual layer thickness data" P8L27-28 – I don't understand this statement. A step change in accumulation rate is similar to a DO event. Ice sheets are able to handle this just fine. So what are you talking about? P8L29-31 – I am still confused. Why is a piece-wise parabolic annual layer thickness model equivalent to a piece-waise linear vertical strain rate or parabolic annual accumulation rate? Isn't this only true for a steady-state? I have no idea what point you are trying to make it. Please articulate it more clearly. P8L31-P9L1 – If this is similar to ALT (Fudge et al., 2014) except with greater discontinuity in layer thicknesses which you try to minimize, why not just ALT? You need to justify why your piece-wise parabolic method is applicable. P9L1-5 – Wait? The piece-wise parabolic method can't be used at the 8.2k age tie and requires some convoluted correction? OK, you really need to justify why this is a worthwhile approach. It seems ill-suited. P10-L1-4 – Why should the variation in annual layer thickness be described by a quadratic? I realize you can do this mathematically, but why? Particularly when you have to add corrections like above. P10-L10 – what are the corrections? This seems to be getting overly complicated. P10-L1-16 – This whole description needs some figures to help explain it. I'm thoroughly lost (why do you use only a subset of age ties to avoid overfitting) which I think is primarily because you haven't taken the effort to really explain this. P10L29 – "some of the random ensemble members have the order of age ties swapped and must be rejected". This means your uncertainties are correlated. Thus, all of your uncertainty estimates from this process will be underestimated. You can't just throw out the reversed order age ties. You need to re-evaluate your age ties and your method. P10L30-P11-2 – What is the standard deviation for the annual layer thickness? It is not given in 3.5. Further, I don't think this captures the full uncertainty since the interval in question is 2x longer than that used to develop the uncertainty. The interval used may have missed important variations in

climate that affect the accumulation rate and hence the layer thicknesses. P11L4-5 – This sentence raises so many red flags. First, "robust" seems a little strong when you are not accounting for 1) dependence in the uncertainties of many of the age ties, 2) the uncertainty in the AICC; and 3) the variability in annual layer thicknesses between tie points. Second, could you help the reader out and explain that you are using a median absolute deviation calculation rather than making us guess at it? Also explain why using a 2 standard deviation (say 3% and 97% percentiles ) is not as good. Figure 1 – The age model does not depend on the skewness of the annual layer tie points. There are so many more important figures that could be included. P11L10 – be clear what you mean. The change in slope does not necessarily imply a change in the current vertical strain rate. It may imply a past change in the vertical strain. This is another instance where it seems like the thinking about layer thinning and annual layer thicknesses assumes steady-state conditions.

Figure 2 – Add the bed. You focus on the change in slope at ∼8ka, but this figure accentuates the change in slope at ∼30ka, where the slope becomes roughly linear, implying a constant annual thickness. This figure would benefit greatly from a plot of the annual layer thickness. The near-constant annual layer thickness in the deep ice could be due to either stagnant ice or due to basal melt. The authors exclude basal melt, but I'm not sure this is wise, particularly for the glacial period. The annual layer thickness near the bed should approximately match the basal melt rate. The annual layer thickness ∼5e-4m (half a millimeter). So if this melt rate persisted for 50,000 years, 25m of ice would be melted off. Does the ice core really exclude a period of mild melting like this? The authors should show the dating of the proposed the previous interglacial ice and then use a transient thermomechanical model to determine what scenarios can actually be excluded.

P11L21 – why do you limit yourself to just the new section of the annual layer count. You could define tie points throughout the upper core and test a much larger section.

Figure 3 – plot this as age uncertainty, which is more interpretable than the depth

uncertainty.

P12L1-P13L2 – The test period of the accuracy of the uncertainty is too short (∼400 years). The method needs to be tested both for the upper part of the Law Dome timescale and for the WAIS Divide ice core. Instead of ∼2000 years of an annual layer interpretation, there are ∼30,000 years. This encompasses a much wider range of climate variations and will provide a much more accurate assessment of the uncertainty. P14L3 – You need to present the data. If you make 15 matches, the every other one, of course you are going to get good agreement on the remaining ones. But that doesn't tell you anything about whether the ages are actually correct. P14L13 – "differs slightly"? According to Figure 5, LD2017 and LD1 differ by more than 1000 years, which is approximately 3 times the stated uncertainty in LD1 (to the extent this can be evaluated since LD1 was based on an undisclosed dust tie point at the onset of deglaciation). The primary purpose of the LD1 timescale was to assess centennial scale phasing. To be off by a millennia is not "slightly". P14L13-P15L5 – The understatement continues. The age difference at four of the six events in Pedro et al. (2011) Table 4 differ by more than the stated uncertainty (which is exaggerated for the purpose of this comparison because it includes the statistical uncertainty). To write that LD2017 "closely follows" LD1 is incredibly misleading and ignores the context in which LD1 was used. It was used to examine centennial scale timing, so being changed by multiple centuries is a major change. Change the dating of the 80ka ice by multiple millennia is not big deal since no claims about the accuracy of the age scale at those depths have really been made. But to change the timing of deglacial events by centuries is because of how Law Dome has been used in the past. P15L7-9 – These sentences reveal a complete lack of understanding of timescale uncertainties and a fundamental laziness in the analysis. First, the lack of understanding: the authors compare an uncertainty – which is relative to the AICC – to an uncertainty which is not relative to the AICC. The timing of these events is not directly comparable and in a manuscript about "robust uncertainties" whose primary goal is to clean-up the mess of the Law Dome timescale, this is unacceptable. Second, the laziness: the authors use the exact num-

bers calculated by Pedro et al. despite 1) that number being computed with Law Dome isotope record on the inaccurate LD1 timescale, 2) the large uncertainties identified by Fudge et al. for both the Siple Dome and Byrd timescales and 3) the new information from cores such as WAIS Divide. This was so appalling I had to go get another cup of coffee to even face it. P15L12 – stop using "slightly". These are important differences. P15L21 – "we show good agreement" – No, you do not! Figure 6 shows that even with a gigantic accumulation range, the old accumulation rates exceed that range 1/3 of the time. How is that good? Further, with accumulation ranges this large you are basically spanning all available accumulation rates. If someone as how much rain Hobart gets and you answered somewhere between 0.2 and 1.5 meters, would that be helpful? You've described anything from a desert to a rain forest. Lest you think this comparison irrelevant, the range I describe is 2 times the range in Figure 6. P16L7 – The comparison to central plateau sites is not useful. You write as much. Delete his paragraph. P17L3 – Wait, you have d15N of N2? Why don't you use it to constrain your firn modeling? I can't tell if you are using this data to support your lower bound and accepting that it contradicts your lower bound. This needs to be integrated into your analysis. P17L7-12 – The dust concentration is not useful given the uncertainties and likely is controlled by processes other than the amount of precipitation at the site. Just delete it. P17L13 – What is the purpose of this paragraph? It has a lot of speculation about ice flow in the deepest ice, but leads nowhere. I'm not convinced that the bed topography is particularly important, in part because the authors don't present any data on the bed topography. Figure 7 – support your statement "due to the influence of bed undulations". So you see a different velocity profile in areas with a different bed topography? This seems speculative to me and not backed up by data. Figure 7 – why is there no data between ∼70 and 100m above the bed? Is there a measurement problem? Where does the dusty glacial ice start? What is the fabric like? Do you have a false bed situation like Siple Dome?

---

## Referee Comment (RC3) · Anonymous Referee #3 · 22 Sep 2017

Roberts and colleagues describe a new ice age scale for the Law Dome ice core in Antarctica, that is based on a combination of layer counting and stratigraphic matching based on volcanic deposits, CH4 d18O-atm and d18O-ice. The combination of these different approaches is somewhat ad-hoc, but I understand age scale construction is a pragmatic endeavor and one must make use of what is available.

Clim. Past has a history of publishing ice core time scales, and therefore the manuscript fits well within the scope of the journal. However, the work has some clear shortcomings that need to be addressed.

The authors do not show any of the data used in constructing the age scale. Figures need to be included, so readers can judge for themselves whether the CH4, volcanic and d18O matches are valid. LD records should be plotted with the target records.

While the authors describe the Delta-age modeling in great detail, they do not reconstruct it for the entire core, and consequently there is no gas age scale. This should be included, for LD2017 to be a complete chronology. Since the modeling has already been done, this will be trivial to add. Having a gas age scale is further crucial for validating the CH4 and d18O-atm matching described in the text.

I would like the authors to consider converting the time scale from B2k to BP, with present taken to be 1950. The latter is the convention in nearly all fields of paleoclimate (14C dating, U/Th dating, paleoceanography, AICC2012, WD2014, etc). B2k is an unfortunate choice used solely in the GICC05 chronology, that has caused nothing but confusion without being an improvement in any way.

The authors match the ice-ties to AICC2012, and the gas-ties to GICC05. This is potentially very problematic, because these two chronologies are of course not perfectly synchronized (due to the large Antarctic Delta-age). The uncertainty in the AICC2012-GICC05 synchronization is easily several hundred years, i.e. larger than the stated Delta-age.

I have several comments regarding Delta-age:

*The delta-age calculation uses very unorthodox approach to estimating past accumulation, by using the notion that past basal temperatures must remain below the melting point. However, they use a steady-state approach, which is far from perfect given that the LGM minimum only lasted for a short duration. Also, the ice thickness is far from certain, as there are virtually no data-based constraints.

*It is unclear what the Delta-age in Table 2 is based on – it appears to be the higher value based on vapor pressure. We know this is a poor estimate in coastal locations due to large contributions from storm systems. *Why is the Delta-age so much smaller than previous estimates by Pedro et al?

*How does the modeled firn column thickness agree with the d15N data shown by

[Figure]

Landais et al. 2006? That would be the better constraint on the problem than the unconventional ones used here.

*The authors can actually test the validity of the Delta age around 15ka, where they have a gas-based and volcanic-based tie in close proximity. Are both consistent?

The stated age uncertainty is much too small for a core that has so few age constraints. At the very least, the AICC2012 uncertainty should be added (in quadrature) to the LD2017 uncertainty, given that those uncertainties propagate into LD2017.

Why are the upper 760 m not included in the LD2017 chronology given in the supplement?

The timing of climatic variations seems to worsen using the LD2017 chronolgy. In the pedro et al. chronology deglacial warming started around 17.8ka, like the rest of Antarctica, whereas in LD2017 it is 18.8ka, which is much too early. However, it is hard for the reader to evaluate the agreement with other records, since no d18O data are shown.

Throughout the MS the authors refer to "age-ties", which is an incorrect concept, in my view. What they do is to derive stratigraphic ties, either in the gas or ice phase, after which they use another age model to assign an age to these stratigraphic markers. The tie is not to an age, but to another record. When the EDC chronology changes, the LD2017 chronology should change along with it.

The method of piecewise parabolic fitting is not at all well described. It was not clear to me how they ensure continuity over the age points. Also, how are the fitting parameters a,b and c found? The difference with the smooth layer method of Fudge et al. is not well justified.

It is not clear what the implications are for the last glacial climate that are promised in the title

---

## Author Comment (AC1) · 6 Nov 2017

We thank the reviewers for their valuable reviews of this manuscript and constructive suggestions. We will be revising the work accordingly and have no doubt that this will assist us in making significant improvements to the work. Due to extended Antarctic field commitments this season this revision cannot be completed soon and we will therefore not submit a revision at this time. We will work with Climate of the Past to submit again as soon as possible.

Tas van Ommen

(On behalf of Jason Roberts and co-authors).